# Algorithm Research Using GNSS-TEC Data to Calibrate TEC Calculated by the IRI-2016 Model over China

**Wen Zhang** [1,2] , **Xingliang Huo** [1,2,*], **Yunbin Yuan** [1,2], **Zishen Li** [3] **and Ningbo Wang** [3]

1   State Laboratory of Geodesy and Earth's Dynamics, Innovation Academy for Precision Measurement Science and Technology, Chinese Academy of Sciences, Wuhan 430077, China; zhangwen@apm.ac.cn (W.Z.); yybgps@asch.whigg.ac.cn (Y.Y.)
2   University of Chinese Academy of Sciences, Beijing 100049, China
3   Aerospace Information Research Institute, Chinese Academy of Sciences, Beijing 100094, China; lizishen@aircas.ac.cn (Z.L.); wangningbo@aoe.ac.cn (N.W.)
*   Correspondence: xlhuo@apm.ac.cn

**Abstract:** The International Reference Ionosphere (IRI) is an empirical model widely used to describe ionospheric characteristics. In the previous research, high-precision total ionospheric electron content (TEC) data derived from global navigation satellite system (GNSS) data were used to adjust the ionospheric global index $IG_{12}$ used as a driving parameter in the standard IRI model; thus, the errors between IRI-TEC and GNSS-TEC were minimized, and IRI-TEC was calibrated by modifying IRI with the updated $IG_{12}$ index (IG-up). This paper investigates various interpolation strategies for IG-up values calculated from GNSS reference stations and the calibrated TEC accuracy achieved using the modified IRI-2016 model with the interpolated IG-up values as driving parameters. Experimental results from 2015 and 2019 show that interpolating IG-up with a 2.5° × 5° spatial grid and a 1-h time resolution drives IRI-2016 to generate ionospheric TEC values consistent with GNSS-TEC. For 2015 and 2019, the mean absolute error (MAE) of the modified IRI-TEC is improved by 78.57% and 77.42%, respectively, and the root mean square error (RMSE) is improved by 78.79% and 77.14%, respectively. The corresponding correlations of the linear regression between GNSS-TEC and the modified IRI-TEC are 0.986 and 0.966, more than 0.2 higher than with the standard IRI-TEC.

**Keywords:** IRI-2016; TEC; ionosphere; GNSS

## 1. Introduction

The ionosphere is a part of the atmosphere that exists in an ionized state within a range of tens of kilometers to thousands of kilometers from the ground. It is an important part of the geospatial environment and has a significant impact on radio information systems such as global navigation satellite systems (GNSS), communication systems, and radar [1–6]. The International Reference Ionosphere (IRI) is a widely used empirical ionospheric model. It was jointly sponsored by the Committee on Space Research (COSPAR) and the International Union of Radio Science (URSI) and is based on a large number of observation data (including ionosonde, incoherent scattering radar, sounding rocket, and satellite data), which provide important environmental parameters on a monthly basis, such as the total ionospheric electron content (TEC), electron density profiles, the peak electron density (NmF2), and the peak electron density height (hmF2), as well as several additional parameters that are used to describe the monthly changes in the ionospheric state [7–11]. As an empirical model, IRI is updated on the basis of an abundance of observed ionospheric data and the continuous optimization of the parameter calculation model. At present, IRI-2016 is the latest version of the IRI model. The most important change made in this version is the addition of two models for calculating the peak electron density height in the F2 layer: the AMTB and Shubin models [12]. Similar to other versions of the IRI model, the IRI-2016 model uses spherical harmonic Legendre functions to represent geographical

coordinates; the diurnal and seasonal variations in the ionospheric critical frequency of the F2 layer (foF2) and the peak electron density and height are described by CCIR and URSI coefficients, respectively; the 12-month average global ionospheric index, $IG_{12}$, is used as the main driving control parameter of the model to calculate the peak electron density, NmF2; and the 12-month global average number of sunspots, $RZ_{12}$, is used as the main driving control parameter of the model to calculate the peak electron density height, hmF2 [13].

The ionosphere is mainly affected by solar radiation and the Earth's atmosphere, and there are regular changes, but there will be some abnormal phenomena due to the interference of other factors. Common ionospheric anomalies include equatorial ionospheric anomalies (EIA), equatorial plasma bubbles (EPBs), polar tongue of ionization (TOI), ionospheric "noontime bite-out" phenomenon, etc. Theoretically, the electron density should be the highest in the equatorial region during the day. However, under the combined action of electric field (E), magnetic field (B), gravity and pressure gradient forces, the peak value of electron density occurs at the magnetic latitude $\pm15$, namely, EIA [14,15]. The EPBs is a kind of nighttime plasma irregularity that occurs within the equatorial and low-latitude region due to the horizontal direction of the magnetic field lines in this region [16,17]. The TOI is a continuous and dense current along the global convective pattern, which is a limited region of increased plasma density in the polar cap region [18,19]. However, the IRI model cannot always accurately reflect these ionospheric abnormal phenomena.

Over time, different groups have attempted to use various methods to improve the accuracy of the IRI model to improve its applicability in scientific research and engineering practice. Using GNSS data to extract high-precision ionospheric TEC [20–30] and improve the IRI model based on GNSS TEC is an important approach. The corresponding methods can be mainly divided into two categories. Methods in the first category use the data assimilation technique, taking the IRI model as the background ionosphere and GNSS-TEC data as observation data, and thereby obtain more reasonable and credible results [31–34]. For example, ErCha et al., using IRI as the background model and GNSS data as the observation values, applied a three-dimensional variational method and the Kalman filter to assimilate the ionospheric data, and generated quasi-real-time predictions of ionospheric TEC over China and adjacent areas [32]. Methods in the second category rely on the ingestion of GNSS data to minimize the difference between the high-precision TEC values extracted from GNSS data and the TEC results output from the IRI model by adjusting the $IG_{12}$ index and the $RZ_{12}$ index to improve the model's accuracy [35–41]. For instance, Nicholas Ssessanga et al. ingested GNSS-TEC data into the IRI-2012 model, and by adjusting the $IG_{12}$ and $RZ_{12}$ indices simultaneously, obtained a modified IRI-2012 model that was more accurate than the original model in estimating TEC [39]. Lei Liu et al. incorporated global ionosphere map (GIM) TEC data from Europe into IRI-2016 and retrieved the effective ionospheric index per hour at different latitudes to improve the accuracy of the IRI model [41]. Notably, in this method, the updated $IG_{12}/RZ_{12}$ index is a parameter that includes the error of the CCIR/URSI coefficient rather than a means of characterizing the original sunspot and ionospheric variation activities.

The $IG_{12}$ index is a driving parameter of IRI model which was introduced by Liu et al. It is obtained by adjusting the CCIR model of foF2 to the noontime measurements of several reference ionosonde stations [42]. At present, the index is produced based on four stations (two from the Northern Hemisphere and two from the Southern Hemisphere), which limits the reliability of this index to represent the global ionospheric conditions. This paper mainly focuses on the second approach to improve the IRI-2016 model based on GNSS-TEC data, namely updating the $IG_{12}$ index with GNSS-TEC data to improve the accuracy of the TEC values calculated by the IRI-2016 model. Importantly, the updated values of the $IG_{12}$ index based on GNSS-TEC are different in different regions and at different times. Therefore, the interpolation of the updated $IG_{12}$ for engineering users without GNSS-TEC to obtain the calibrated TEC from the modified IRI-2016 model, such as users of navigation and positioning or users of radar communication, is important for reducing the impact of

the ionosphere on such systems. Consequently, this paper proposes various methods of interpolating the updated values of the $IG_{12}$ index in time and space and will demonstrate and evaluate the impact of these different interpolation methods on the accuracy of the TEC values over China calculated using the IRI-2016 model.

In addition, because the ionospheric electron density derived from the IRI model is related to changes in the ionization layer thickness parameters, peak electron density height, and other factors, using GNSS-TEC data to improve the accuracy of the electron density calculated by the IRI model is a relatively complex task. Therefore, this paper does not discuss electron density accuracy; moreover, given that the Shubin model performs better than the AMTB model over China [43], the Shubin model is used in this study to calculate the peak electron density height (hmF2) rather than the AMTB model, which is the default selection in the IRI model. Because the Shubin model does not involve the $RZ_{12}$ index, we do not update the $RZ_{12}$ index.

In summary, the framework of this paper is as follows. Section 1 is the introduction, Section 2 describes the experimental data and experimental methods, Section 3 presents and discusses the experimental results, and Section 4 presents the conclusion of this paper.

## 2. Data and Methodology

### 2.1. Data

The Fortran code version of the IRI-2016 model (Available online: http://irimodel.org, accessed on 2 May 2019) was used in this analysis. The GNSS-TEC data used were accurately extracted from the observation data of 58 GNSS receiver stations distributed relatively uniformly over China. The inversion of GNSS-TEC data has been illustrated in several previous papers [44–46]. The TEC data obtained from 46 of the GNSS receiver stations were used to iteratively update the $IG_{12}$ index of the IRI-2016 model, and the TEC data obtained from the remaining 12 stations were used to evaluate the accuracy of the TEC results calculated using the improved IRI-2016 model. The locations of the GNSS receiver stations are shown in Figure 1. Considering the close relationship between the ionosphere and solar activity [47,48], we chose data spanning one day a week in 2015 and 2019 to include the variations occurring during high solar activity and low solar activity; moreover, considering the impact of geomagnetic activity on the ionospheric TEC, only dates when the disturbance storm time (Dst) index was greater than −30 nt, namely, quiet days, were selected in this study.

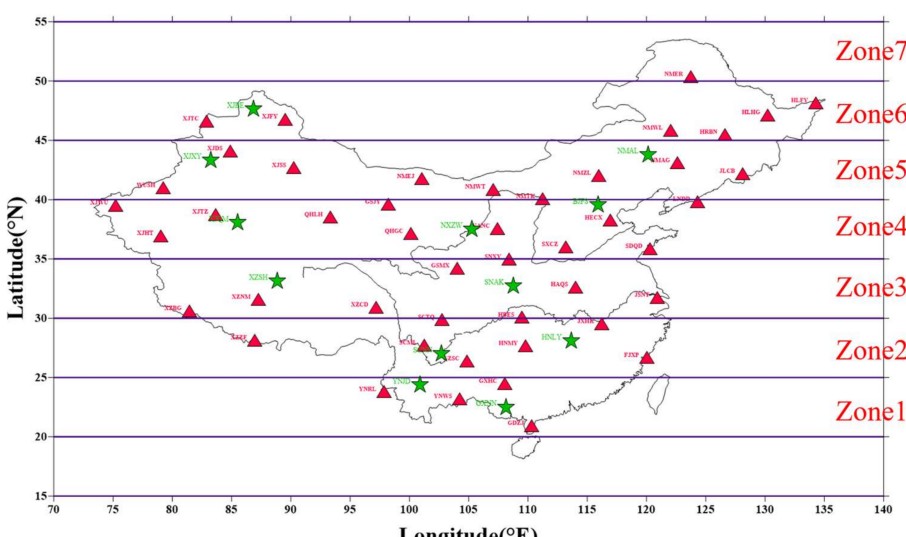

**Figure 1.** Distribution of the 58 GNSS stations used in this work (the red triangles represent the stations used to improve the IRI-2016 model, and the green five-pointed stars represent the stations used to evaluate the improvement effect on the IRI-2016 model).

*2.2. Methodology*

Referring mainly to the work of Ssessanga N et al. [39], this paper presents a method in which high-precision GNSS-TEC data are used to drive the IRI-2016 model to adjust the $IG_{12}$ index to optimize the model performance. Afterward, the updated $IG_{12}$ index (IG-up) is interpolated using various temporal and spatial interpolation methods to obtain spatiotemporally continuous and high-precision IRI-TEC values to meet the needs of users in China. First, to obtain the IG-up values, the difference (DTEC) between GNSS-TEC as estimated from the observation data of GNSS stations at 46 different locations and IRI-2016-TEC is used to iteratively adjust the $IG_{12}$ index, such that DTEC is below a set threshold ( $|DTEC| < 0.5$ TECu). To some extent, the IG-up values reflect the error of the CCIR coefficient in describing foF2 at different times and spatial locations; consequently, IG-up varies greatly with time and space. Taking the first four days of 2015 as an example, Figure 2 shows the changes in time and space between the original $IG_{12}$ index of the IRI-2016 model and IG-up at stations HRBN (45.70°N, 126.62°E) and GDZJ (21.15°N, 110.30°E). On this basis, this section will discuss various interpolation methods for IG-up values at different temporal and spatial scales and evaluate the accuracy of the TEC results estimated by the IRI-2016 model when driven by the interpolated IG-up values.

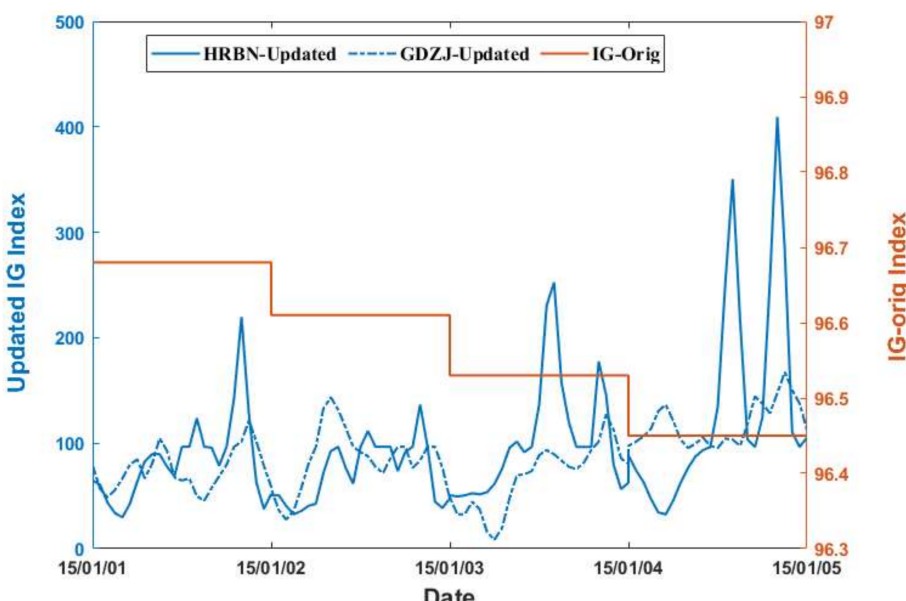

**Figure 2.** Updated $IG_{12}$ index (IG-up) at stations HRBN and GDZJ (HRBN-Updated and GDZJ-Updated) and the original $IG_{12}$ index (IG-Orig).

In Figure 2, the original $IG_{12}$ values of the two stations are directly obtained from the internal ig_rz.dat in the IRI model, and the updated $IG_{12}$ values are calculated iteratively by ingesting GNSS-TEC data into the IRI-2016 model. The difference between IRI-TEC and GNSS-TEC determines the changing trend of the updated $IG_{12}$. The updated $IG_{12}$ index (IG-up value) varies greatly with time, as shown in Figure 2. Using different time intervals when designing an IG-up interpolation scheme will affect the accuracy of TEC estimation using the improved IRI-2016 model. Thus, this paper designs and compares three different temporal interpolation schemes for IG-up:

Scheme 1: The $IG_{12}$ index is iteratively updated at time intervals of 1 h, that is, at 0:00, 1:00, 2:00, ..., 24:00, to obtain the IG-up value for each hour.

Scheme 2: The $IG_{12}$ index is iteratively updated at time intervals of 2 h, that is, at 0:00, 2:00, 4:00, ..., 24:00, to obtain the IG-up value every two hours.

Scheme 3: The $IG_{12}$ index is iteratively updated at time intervals of 4 h, that is, at 0:00, 4:00, 8:00, ..., 24:00, to obtain the IG-up value every four hours.

Expression (1) is used to calculate the interpolated IG-up $IG(t)$ for each minute in accordance with the above three schemes, and the interpolation effects of the three time-interval schemes are then compared:

$$IG(t) = \frac{T_{i+1} - t}{T_{i+1} - T_i} IG_i + \frac{t - T_i}{T_{i+1} - T_i} IG_{i+1}, \tag{1}$$

where $T_i$ and $T_{i+1}$ are two consecutive epochs and $IG_i$ and $IG_{i+1}$ are the IG-up values corresponding to these two epochs, respectively.

Figure 2 shows that there are also obvious differences in the IG-up values in different regions. Therefore, it is also necessary to investigate the impact of the spatial interpolation scheme for IG-up on the accuracy with which the IRI-2016 model is driven to calculate the TEC. In this paper, two spatial interpolation schemes are designed and discussed.

Scheme A: Considering that the ionosphere is greatly affected by latitude, the China region (20–55°N, 70–135°E) is divided into seven latitudinal zones, each with a 5° latitudinal span. The IG-up values at the GNSS stations included in each latitudinal zone are averaged, and the result is then used as the effective IG-up value everywhere in the corresponding latitudinal zone. The latitudinal zone divisions are shown in Figure 1.

Scheme B: The China region is divided into a grid with a spatial resolution of 2.5° in latitude and 5° in longitude, and the IG-up values at the 46 GNSS stations are used to assign values to the grid nodes in accordance with the inverse distance weighting method to obtain an IG-up map. Then, the effective IG-up value at a specific location can be interpolated on this basis. The distance limit in the inverse distance weighting method is set to 500 km. The applied partitioning method ensures that at least two to six GNSS stations can provide an IG-up value for each grid node on the Chinese mainland. Expression (2) shows the inverse distance weighting method for the IG-up values:

$$IG(\lambda, \beta) = \frac{\sum\limits_{i=1}^{n} \frac{IG_i}{d_i^k}}{\sum\limits_{i=1}^{n} \frac{1}{d_i^k}}, \tag{2}$$

where $IG(\lambda, \beta)$ is the effective IG-up value at the grid node $(\lambda, \beta)$, $IG_i$ is the IG-up value at the $i$-th GNSS observation station surrounding the grid node, and $d_i$ is the spherical distance from this observation station to the grid node. $k$ is the power of the inverse distance; generally, $0 \leq k \leq 3$. The larger the value of $k$ is, the more prominent the role of adjacent points [49,50]; in this study, $k = 2$. After obtaining the grid map of the IG-up values, the user can refer to the interpolation method for TEC products in the International GNSS Service (IGS) IONEX format [51] and use Expression (3) to interpolate IG-up:

$$IG(\lambda_0 + p\Delta\lambda, \beta_0 + q\Delta\beta) \\ = (1-p)(1-q)IG_{0,0} + p(1-q)IG_{1,0} + q(1-p)IG_{0,1} + pqIG_{1,1}, \tag{3}$$

where $IG(\lambda_0 + p\Delta\lambda, \beta_0 + q\Delta\beta)$ is the effective IG-up value at any position to be interpolated; $IG_{0,0}$, $IG_{0,1}$, $IG_{1,0}$, and $IG_{1,1}$ are the IG-up values at the four grid points closest to a specific interpolation point; $p$ and $q$ represent the distances along with the longitudinal and latitudinal directions, respectively, between the point to be interpolated and $IG_{0,0}$, satisfying $0 \leq p < 1$ and $0 \leq q < 1$; $\Delta\lambda$ and $\Delta\beta$ are the longitudinal and latitudinal intervals of the grid, respectively.

### 2.3. Evaluation Methodology

1. Evaluation scheme for the temporal interpolation of IG-up: High-precision TEC data extracted from six GNSS stations at different latitudes are used to drive the IRI-2016 model to calculate the IG-up values at different integer hours, and the $IG(t)$ corresponding to a 1-min sampling interval is calculated via interpolation under scheme 1, scheme 2 and scheme 3 using Expression (1). Then, the interpolated results

are substituted back into the IRI-2016 model to drive the output TEC. On this basis, we calculate the mean absolute error (MAE), root mean square error (RMSE), and precision improvement (PI) of the TEC estimates obtained with different time-interval schemes using Expression (4), Expression (5), and Expression (6), respectively, to evaluate the interpolation effects of the different time-interval schemes for IG-up.

2. Evaluation scheme for the spatial interpolation of IG-up: High-precision TEC data extracted from 46 GNSS stations at different latitudes are used to drive the IRI-2016 model to calculate the IG-up values at different integer hours, and Expression (2) is used to calculate an IG-up map with a spatial resolution of 2.5° in latitude and 5° in longitude within the latitudinal range of 20°N–55°N and the longitudinal range of 70°E–135°E. On this basis, the IG-up values at integer hours corresponding to 12 other GNSS stations at different latitudes are then calculated via interpolation using Expression (3). At the same time, the average value of IG-up in each latitudinal zone is calculated, and the results are then substituted back into the IRI-2016 model to drive the output TEC. The two-dimensional (2-D) distribution of the calculated TEC output is compared with that of GNSS-TEC in China, the difference (DTEC) between the calculated TEC output and GNSS-TEC is calculated, and the DTEC distributions are compared using boxplots to evaluate the effects of different spatial interpolation schemes for IG-up.

3. Evaluation scheme for the integrated interpolation of IG-up in time and space: Using high-precision TEC data extracted from 12 GNSS stations at different latitudes as a reference, the IG-up values obtained using the integrated interpolation scheme are substituted back into the IRI-2016 model to drive the output TEC. Then, the accuracy indices MAE, RMSE, and PI are calculated using Expression (4), Expression (5) and Expression (6), respectively, and the linear regression correlations of the TEC values are analyzed:

$$\text{MAE} = \frac{1}{n} \sum_{k=1}^{n} |TECG(k) - TECiri(k)|, \tag{4}$$

$$\text{RMSE} = \sqrt{\frac{\sum_{k=1}^{n} |TECG(k) - TECiri(k)|^2}{n}}, \tag{5}$$

$$\text{PI} = \frac{X_{UPDATE} - X_{ORIGIN}}{X_{ORIGIN}} \cdot 100[\%], \tag{6}$$

where $TECG(k)$ represents GNSS-TEC, $TECGiri(k)$ is the TEC output before and after improvement with various schemes, $k$ is the sample index, $X_{ORIGIN}$ and $X_{UPDATE}$ represent the MAE or RMSE of the TEC results calculated using the IRI-2016 model before and after improvement, respectively. Accordingly, PI represents the precision improvement corresponding to either the MAE or RMSE.

## 3. Results and Analysis

### 3.1. Comparison of IG-Up Interpolation Schemes with Different Time Intervals

Considering that the ionospheric TEC is greatly affected by solar activity and latitude [52,53], this paper uses data from six GNSS stations distributed relatively uniformly at different latitudes and ensures that there are enough available data on these six stations, HLHG (47.4°N/130.2°E), NMAG (43.3°N/122.6°E), NMTK (40.2°N/111.3°E), SNXY (34.2°N/108.4°E), XZNM (31.8°N/87.2°E), and GXHC (24.7°N/108.1°E), to calculate IG-up values in accordance with the three time interpolation schemes introduced in Section 2.2 for 2015 and 2019, which are then interpolated and plugged back into the IRI-2016 model to drive the calculation of TEC time series with 1-min time intervals for comparison with the high-precision GNSS-TEC results; see Section 2.3 for the evaluation method.

For example, Figures 3 and 4 compare the TEC estimates obtained using the various schemes and GNSS-TEC on the first day of 2015 and 2019, respectively. It can be seen that the diurnal variation of the TEC estimated by IRI-2016 (IRI-TEC) is significantly different

from that of GNSS-TEC, especially in the daytime, with a maximum difference of 20 TECu, and that IRI-TEC is more different from GNSS-TEC at middle latitudes than at low latitudes over China. GNSS-TEC reflects the ionospheric "noontime bite-out" phenomenon in the daytime; that is, the ionospheric TEC decreases significantly at noon and has a double peak characteristic before and after noon [54,55]. In contrast, the IRI-2016 model can only reflect the diurnal trend of variation. However, when the GNSS-TEC data are used to adjust and update the $IG_{12}$ index and the results are substituted back into the IRI-2016 model, it can be seen that there is no obvious difference in the TEC values between the improved IRI-TEC and GNSS-TEC. In addition, compared with scheme 3 (4-h intervals), the TEC values under scheme 1 (1-h intervals) and scheme 2 (2-h intervals) exhibit a more accurate TEC "noontime bite-out" phenomenon that is more consistent with the GNSS-TEC behavior. From the data for 8–12 UT at station GXHC in Figure 3 and for 12–16 UT at station NMTK in Figure 4, it can be seen that when the TEC value becomes complex, the TEC results of schemes 2 and 3 exhibit obvious fluctuations compared with GNSS-TEC, while the TEC results of scheme 1 are consistent with GNSS-TEC.

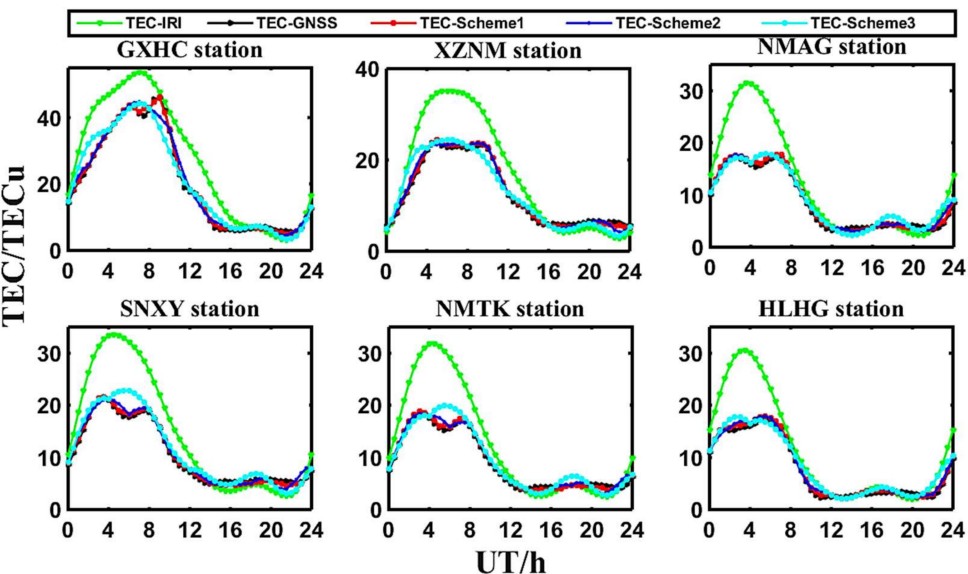

**Figure 3.** Comparison between TEC estimates corrected using different temporal interpolation schemes and GNSS-TEC at six receiver stations on the first day of 2015.

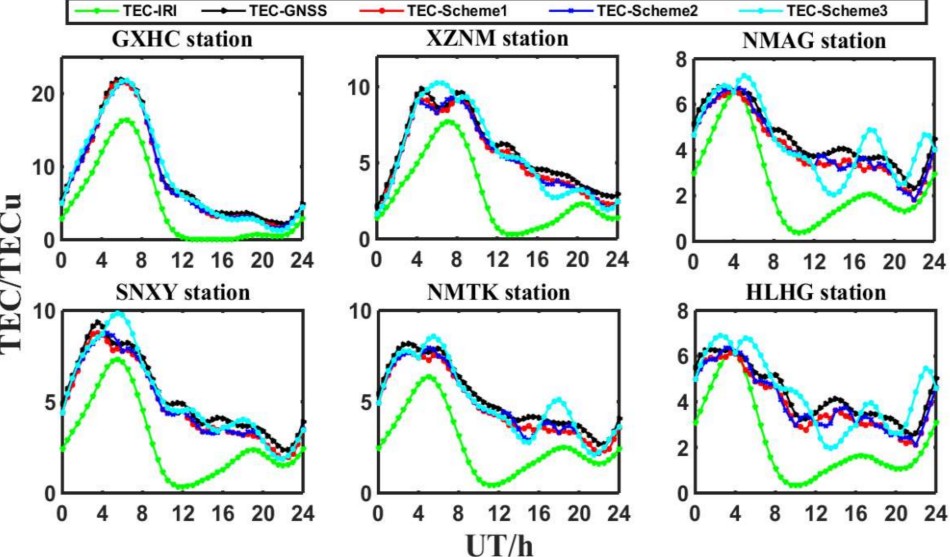

**Figure 4.** Comparison between TEC estimates corrected using different temporal interpolation schemes and GNSS-TEC at six receiver stations on the first day of 2019.

Table 1a through Table 2b record the accuracy indices, including the MAEs, RMSEs, and PIs, between the IRI-TEC calculated using the IRI-2016 model driven by the original $IG_{12}$ values and GNSS-TEC and between the modified IRI-TEC results calculated using the IRI-2016 model driven by the three different IG-up temporal interpolation schemes and GNSS-TEC in 2015 and 2019. The sampling interval used is 1 min.

**Table 1.** (**a**). MAEs and MAE PIs for the TEC in 2015. (**b**). RMSEs and RMSE PIs for the TEC in 2015.

**(a)**

| | MAE (TECu) | | | | MAE PI | | |
|---|---|---|---|---|---|---|---|
| Station | IRI-2016 | Scheme 1 | Scheme 2 | Scheme 3 | Scheme 1 | Scheme 2 | Scheme 3 |
| GXHC | 11.5 | 0.5 | 1 | 2.8 | 95.65% | 91.30% | 75.65% |
| XZNM | 4.2 | 0.5 | 0.7 | 1.3 | 88.10% | 83.33% | 69.05% |
| NMAG | 3.5 | 0.4 | 0.5 | 1.0 | 88.57% | 85.71% | 71.43% |
| SNXY | 3.4 | 0.4 | 0.6 | 1.1 | 88.24% | 82.35% | 67.65% |
| NMTK | 3.5 | 0.4 | 0.5 | 1.0 | 88.57% | 85.71% | 71.43% |
| HLHG | 3.6 | 0.5 | 0.5 | 1.0 | 86.11% | 86.11% | 72.22% |
| Average | 5.0 | 0.5 | 0.6 | 1.4 | 90.00% | 88.00% | 72.00% |

**(b)**

| | RMSE (TECu) | | | | RMSE PI | | |
|---|---|---|---|---|---|---|---|
| Station | IRI-2016 | Scheme 1 | Scheme 2 | Scheme 3 | Scheme 1 | Scheme 2 | Scheme 3 |
| GXHC | 16.3 | 0.6 | 1.5 | 4.2 | 96.32% | 90.80% | 74.23% |
| XZNM | 5.7 | 1.0 | 1.2 | 1.9 | 82.46% | 78.95% | 66.67% |
| NMAG | 4.3 | 0.5 | 0.6 | 1.3 | 88.37% | 86.05% | 69.77% |
| SNXY | 4.5 | 0.5 | 0.8 | 1.5 | 88.89% | 82.22% | 66.67% |
| NMTK | 4.3 | 0.5 | 0.7 | 1.3 | 88.37% | 83.72% | 69.77% |
| HLHG | 4.6 | 0.7 | 0.8 | 1.7 | 84.78% | 82.61% | 63.04% |
| Average | 6.6 | 0.6 | 0.9 | 2.0 | 90.91% | 86.36% | 69.70% |

**Table 2.** (**a**). MAEs and MAE PIs for the TEC in 2019. (**b**). RMSEs and RMSE PIs for the TEC in 2019.

**(a)**

| | MAE (TECu) | | | | MAE PI | | |
|---|---|---|---|---|---|---|---|
| Station | IRI-2016 | Scheme 1 | Scheme 2 | Scheme 3 | Scheme 1 | Scheme 2 | Scheme 3 |
| GXHC | 5.1 | 0.4 | 0.6 | 1.2 | 92.16% | 88.24% | 76.47% |
| XZNM | 3.1 | 0.4 | 0.5 | 0.9 | 87.10% | 83.87% | 70.97% |
| NMAG | 2.0 | 0.4 | 0.4 | 0.7 | 80.00% | 80.00% | 65.00% |
| SNXY | 2.5 | 0.4 | 0.5 | 0.7 | 84.00% | 80.00% | 72.00% |
| NMTK | 2.2 | 0.4 | 0.5 | 0.7 | 81.82% | 77.27% | 68.18% |
| HLHG | 2.2 | 0.4 | 0.5 | 0.8 | 81.82% | 77.27% | 63.64% |
| Average | 2.9 | 0.4 | 0.5 | 0.8 | 86.21% | 82.76% | 72.41% |

**(b)**

| | RMSE (TECu) | | | | RMSE PI | | |
|---|---|---|---|---|---|---|---|
| Station | IRI-2016 | Scheme 1 | Scheme 2 | Scheme 3 | Scheme 1 | Scheme 2 | Scheme 3 |
| GXHC | 7.3 | 0.5 | 0.7 | 1.8 | 93.15% | 90.41% | 75.34% |
| XZNM | 4.2 | 0.5 | 0.6 | 1.3 | 88.10% | 85.71% | 69.05% |
| NMAG | 2.5 | 0.4 | 0.5 | 0.9 | 84.00% | 80.00% | 64.00% |
| SNXY | 3.1 | 0.5 | 0.6 | 0.9 | 83.87% | 80.65% | 70.97% |
| NMTK | 2.7 | 0.5 | 0.5 | 0.9 | 81.48% | 81.48% | 66.67% |
| HLHG | 2.7 | 0.5 | 0.5 | 1.0 | 81.48% | 81.48% | 62.96% |
| Average | 3.8 | 0.5 | 0.6 | 1.1 | 86.84% | 84.21% | 71.05% |

According to Table 1 (2015 statistical results) and Table 2 (2019 statistical results), scheme 1 yields the smallest average MAEs (0.4 TECu and 0.5 TECu, respectively) and average RMSEs (0.5 TECu and 1.0 TECu, respectively) among the various temporal inter-

polation schemes and the original IRI-2016 model, and the corresponding average MAE PIs (86.21% and 90.00%, respectively) and RMSE PIs (86.84% and 90.91%, respectively) are also the largest. It is important to note that the MAEs of the ionospheric TEC results at different GNSS stations and under different solar activities obtained using scheme 1 are no greater than 0.5 TECu, which is consistent with the iterative threshold applied when using GNSS-TEC data to improve the IRI-2016 model. This shows that the error of the ionospheric TEC output after the improvement of the IRI-2016 model using scheme 1 is stable.

In contrast, the MAEs and RMSEs obtained under schemes 2 and 3 are not less than 0.5 TECu. At different latitudes and in years with different levels of solar activity, the MAE variation range under scheme 2 is 0.5–1.0 TECu, and the corresponding RMSE variation range is 0.6–1.5 TECu, whereas the MAE variation range under scheme 3 is 0.7–2.8 TECu, and the corresponding RMSE variation range is 0.9–4.2 TECu. In addition, Tables 1 and 2 show that the MAEs and RMSEs under schemes 2 and 3 gradually increase with decreasing latitude and are greater in the year with high solar activity than in the year with low solar activity. According to the PIs in terms of the MAE and RMSE, the PI of scheme 3 is the smallest, and the PI of scheme 2 is better than that of scheme 3.

Summarizing the results presented in Figures 3 and 4 and Tables 1 and 2, it can be concluded that under all investigated conditions, the MAE of IRI-2016 driven by *IG(t)* interpolated from IG-up values calculated at intervals of one hour can achieve the desired threshold constraint of 0.5 TECu, and the TEC from IRI-2016 driven by *IG(t)* can be well-matched with GNSS-TEC. These results indicate that this is the optimal exponential temporal interpolation scheme for IG-up.

*3.2. Comparison of Spatial Interpolation Schemes for IG-Up*

In this section, we analyze and discuss the two spatial interpolation schemes introduced in Section 2.2, namely, scheme A and scheme B. Figures 5 and 6 show the 2-D distributions of GNSS-TEC, the TEC estimates calculated using the original IRI-2016 model, and the TEC estimates obtained using scheme A and scheme B over the China region at the four time points of 0, 6, 12, and 18 UT on April 9 (DOY099) in 2015 and 2019, where grid-TEC and zone-TEC denote the TEC estimates obtained using schemes A and B, respectively.

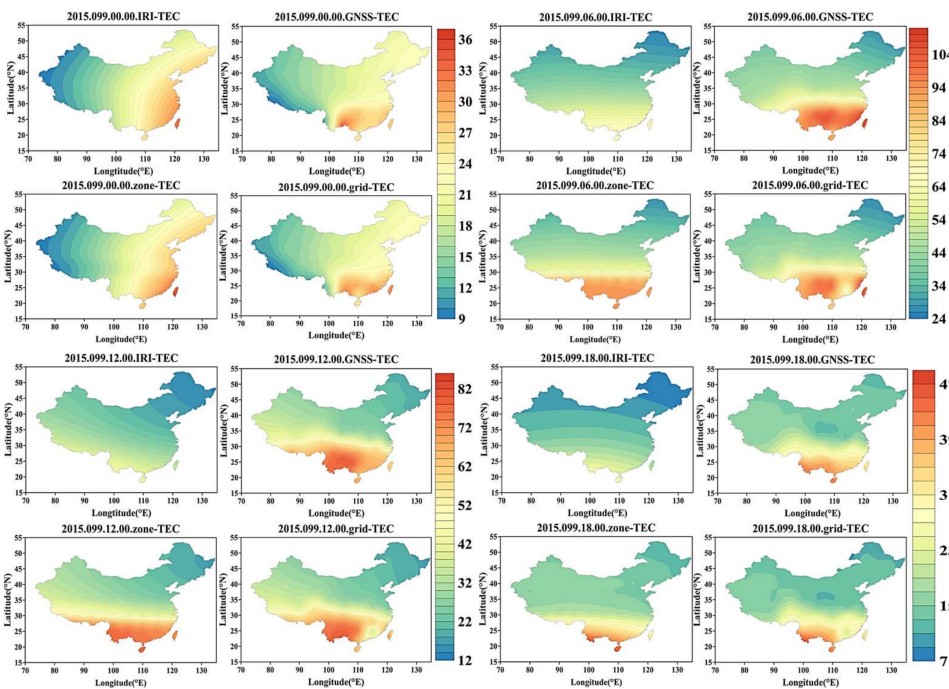

**Figure 5.** 2-D distributions of GNSS-TEC, the TEC estimates calculated using the original IRI-2016 model, and the TEC estimates obtained using schemes A and B in the parts of China region at the four time points of 0, 6, 12, and 18 UT on April 9 (DOY099) 2015.

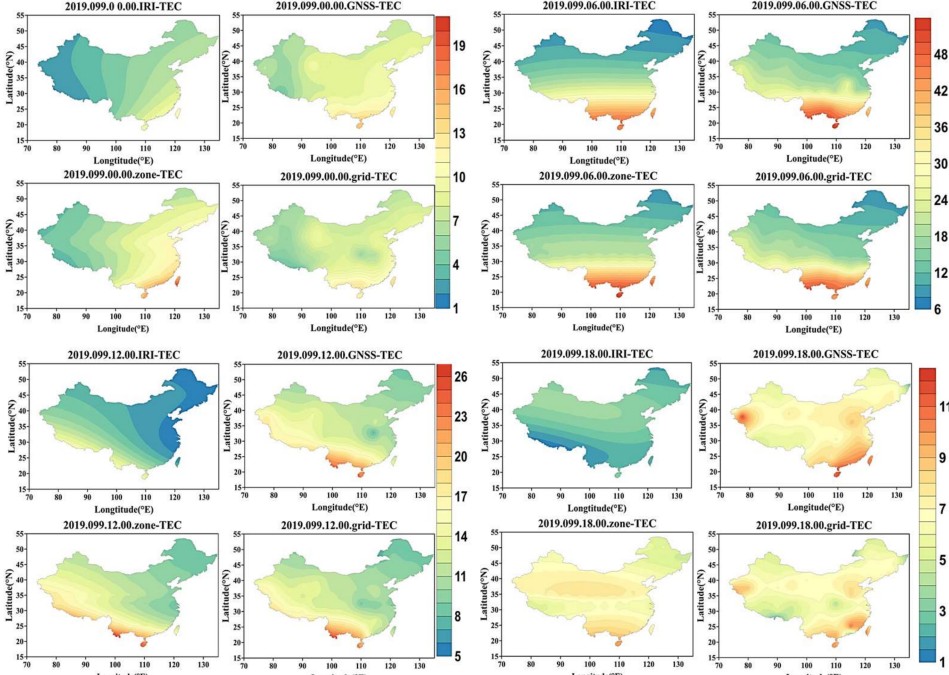

**Figure 6.** 2-D distributions of GNSS-TEC, the TEC estimates calculated using the original IRI-2016 model, and the TEC estimates obtained using schemes A and B in the parts of China region at the four time points of 0, 6, 12, and 18 UT on April 9 (DOY099) in 2019.

From Figures 5 and 6, it can be seen that the ionospheric TEC exhibits obvious characteristics that vary with latitude. Compared to the IRI-2016 model, the 2-D TEC maps over China obtained under schemes A and B are closer to the high-precision GNSS-TEC data, indicating that IRI can be effectively improved by using schemes A and B. However, on a regional scale (20–30° N, 100–120° E), scheme A shows a better improvement effect for

the detailed changes in the TEC than scheme B does. In addition, Figures 5 and 6 show that the TEC estimates obtained via grid-based correction (scheme A) are closest to GNSS-TEC regardless of the solar activity and latitude.

In addition, taking the 12 GNSS monitoring stations at different latitudes (See Figure 1 green five-pointed stars) as examples, this section presents boxplots of the differences (DTEC) between GNSS-TEC and the TEC estimates calculated using IRI-2016, scheme A, and scheme B at integral hours on one day of a week in 2015 and 2019. Each boxplot in Figures 7 and 8 displays the median, maximum, minimum, and upper and lower quartiles for a particular DTEC group. As the latitudes of the GNSS stations (corresponding to the horizontal axis in Figures 7 and 8) gradually increase from left to right, the statistical DTEC results of all schemes gradually decrease. The DTEC results under low solar activity, shown in Figure 8, are smaller than those for the year of high solar activity, shown in Figure 7. The statistical DTEC results of scheme B (blue), in which the IG-up values are averaged by latitude zone, and of scheme A (red), in which the IG-up values are subjected to grid interpolation, are smaller than those of the original IRI-2016 model (green). Figures 7 and 8, combined with the DTEC data statistics, show that the median DTEC results of scheme A at the 12 stations are closer to zero and vary more stable than those of the original IRI-2016 model and scheme B. According to the statistical DTEC results, the median DTEC values under scheme A are closest to zero (0–1.2 TECu for 2015 and −0.2–0.7 TECu for 2019), followed by those under scheme B (−0.9–2.5 TECu for 2015 and −0.1–0.7 TECu for 2019), while those of the original IRI-2016 model are the largest (0.6–6.2 TECu for 2015 and 1.3–2.6 TECu for 2019). In addition, in terms of the upper- and lower-quartile spacings of the DTEC data, the results of scheme A are also more concentrated than those of the other two cases. The upper- and lower-quartile spacings are smallest under scheme A (0.7–3.6 TECu for 2015 and 0.7–1.9 TECu for 2019), second smallest under scheme B (1.9–6.3 TECu for 2015 and 1.2–2.6 TECu for 2019), and largest for the original IRI-2016 model (4.2–15.1 TECu for 2015 and 3.1–6.3 TECu for 2019). Moreover, the maximum and minimum DTEC values under scheme A are significantly better than those under scheme B. The minimum value under scheme A is between −5.9 and −1.1 TECu in 2015 and between −3.7 and −0.6 TECu in 2019, and the maximum is between 1.4 and 8.4 TECu in 2015 and between 1.3 and 3.7 TECu in 2019. The minimum value under scheme B is between −8.9 and −2.7 TECu in 2015 and between −5.3 and −2.0 TECu in 2019, and the maximum value is between 3.3 and 15.8 TECu in 2015 and between 2.8 and 4.9 TECu in 2019. Meanwhile, the minimum value with the original IRI-2016 model is between −6.3 and −18.9 TECu in 2015 and between −1.6 and −11.8 TECu in 2019, and the maximum value is between 10.2 and 35.1 TECu in 2015 and between 6.0 and 13.2 TECu in 2019.

From the above analysis, it can be seen that the grid-based IG-up spatial interpolation with a 2.5° × 5° spatial resolution drives the ionospheric TEC results output by the IRI-2016 model to be closest to GNSS-TEC, indicating that this is the optimal spatial interpolation scheme for the IG-up values.

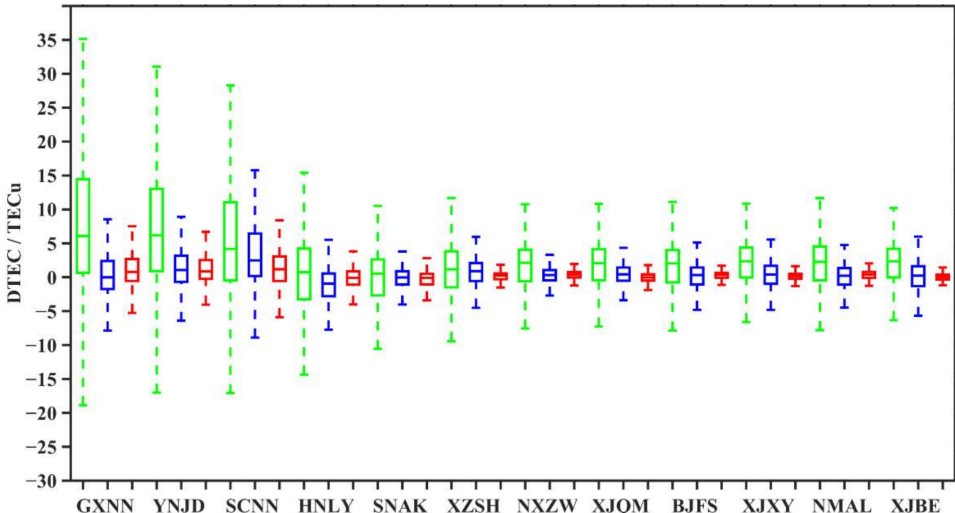

**Figure 7.** Boxplots of the differences (DTEC) between GNSS-TEC and the TEC estimates calculated using the original IRI-2016 model, scheme A and scheme B at 12 verification stations in 2015 (green, blue, and red boxes correspond to IRI-2016, scheme B and scheme A, respectively).

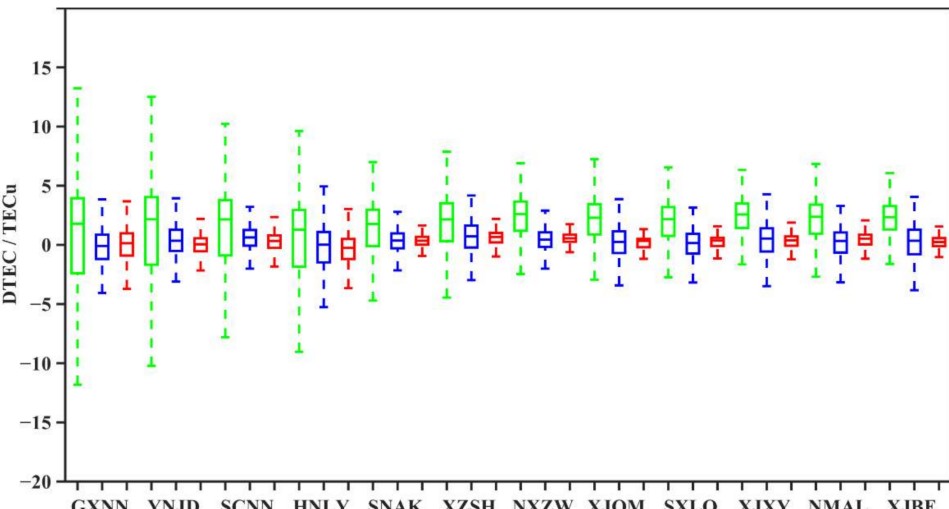

**Figure 8.** Boxplots of the differences (DTEC) between GNSS-TEC and the TEC estimates calculated using the original IRI-2016 model, scheme A and scheme B at 12 verification stations in 2019 (green, blue, and red boxes correspond to IRI-2016, scheme B and scheme A, respectively).

### 3.3. Evaluation of an Integrated Scheme for Interpolating IG-Up in Time and Space

Based on the results discussed in Sections 3.1 and 3.2, this subsection presents hourly IG-up maps calculated by combining scheme 1 and scheme A, with spatial resolutions of 2.5° in latitude and 5° in longitude. Thereupon, the interpolated effective IG-up values are used to drive the IRI-2016 model (called upda-IRI-2016), and the results are statistically evaluated against the high-precision TEC data obtained from 12 GNSS ground stations. The error distribution results are shown in Tables 3 and 4.

**Table 3.** Statistical results for the MAEs, RMSEs, and PIs of the TEC estimates for 2015.

| | MAE (TECu) | | MAE PI | RMSE (TECu) | | RMSE PI |
|---|---|---|---|---|---|---|
| **Station** | **IRI-2016** | **upda-IRI-2016** | | **IRI-2016** | **upda-IRI-2016** | |
| GXNN | 11.2 | 2.6 | 76.79% | 13.6 | 3.2 | 76.47% |
| YNJD | 11 | 2.3 | 79.09% | 13.5 | 2.8 | 79.26% |
| SCNN | 10 | 2.6 | 74.00% | 12 | 3 | 75.00% |
| HNLY | 5.9 | 1.9 | 67.80% | 7.5 | 2.4 | 68.00% |
| SNAK | 3.6 | 1.1 | 69.44% | 4.3 | 1.2 | 72.09% |
| XZSH | 3.7 | 0.5 | 86.49% | 4.5 | 0.6 | 86.67% |
| NXZW | 3.6 | 0.6 | 83.33% | 4.1 | 0.7 | 82.93% |
| XJQM | 3.6 | 0.5 | 86.11% | 4.2 | 0.6 | 85.71% |
| BJFS | 3.6 | 0.5 | 86.11% | 4 | 0.6 | 85.00% |
| XJXY | 3.6 | 0.6 | 83.33% | 4 | 0.7 | 82.50% |
| NMAL | 3.6 | 0.7 | 80.56% | 4.1 | 0.8 | 80.49% |
| XJBE | 3.4 | 0.4 | 88.24% | 3.9 | 0.5 | 87.18% |
| Average | 5.6 | 1.2 | 78.57% | 6.6 | 1.4 | 78.79% |

**Table 4.** Statistical results for the MAEs, RMSEs, and PIs of the TEC estimates for 2019.

| | MAE (TECu) | | MAE PI | RMSE (TECu) | | RMSE PI |
|---|---|---|---|---|---|---|
| **Station** | **IRI-2016** | **upda-IRI-2016** | | **IRI-2016** | **upda-IRI-2016** | |
| GXNN | 4.8 | 1.2 | 75.00% | 5.8 | 1.5 | 74.14% |
| YNJD | 4.6 | 0.7 | 84.78% | 5.5 | 0.9 | 83.64% |
| SCNN | 4.0 | 0.7 | 82.50% | 4.7 | 0.8 | 82.98% |
| HNLY | 3.5 | 1.2 | 65.71% | 4.2 | 1.5 | 64.29% |
| SNAK | 2.6 | 0.5 | 80.77% | 2.9 | 0.6 | 79.31% |
| XZSH | 2.7 | 0.8 | 70.37% | 3.0 | 0.9 | 70.00% |
| NXZW | 2.7 | 0.6 | 77.78% | 3.0 | 0.7 | 76.67% |
| XJQM | 2.4 | 0.4 | 83.33% | 2.7 | 0.5 | 81.48% |
| BJFS | 2.3 | 0.5 | 78.26% | 2.6 | 0.5 | 80.77% |
| XJXY | 2.5 | 0.6 | 76.00% | 2.8 | 0.7 | 75.00% |
| NMAL | 2.4 | 0.7 | 70.83% | 2.7 | 0.8 | 70.37% |
| XJBE | 2.4 | 0.5 | 79.17% | 2.6 | 0.6 | 76.92% |
| Average | 3.1 | 0.7 | 77.42% | 3.5 | 0.8 | 77.14% |

Tables 3 and 4 compare the accuracy indices, including the MAEs, RMSEs, and PIs at 12 GNSS stations, of the IRI-2016 and upda-IRI-2016 models in 2015 and 2019, respectively. Based on the information contained in these tables, the average MAEs of the IRI-2016 and upda-IRI-2016 models at all 12 stations are 5.6 and 1.2 TECu, respectively, in 2015 and 3.1 and 0.7 TECu, respectively, in 2019. The average RMSEs of the IRI-2016 and upda-IRI-2016 models are 6.6 and 1.4 TECu, respectively, in 2015 and 3.5 and 0.8 TECu, respectively, in 2019. The PIs in terms of the MAE and RMSE of the TEC estimates calculated using upda-IRI-2016 relative to those calculated using IRI-2016 are approximately 79% and 77% in 2015 and 2019, respectively. Additionally, since the ionosphere is more active under higher solar activity and at lower latitudes, the statistical MAE and RMSE results gradually increase with decreasing latitude and increasing solar activity. From the results obtained thus far, it seems that calculating hourly IG-up maps for use in the IRI-2016 model can improve the performance of the model when calculating the TEC.

Figures 9 and 10 show the linear relationships between the TEC values calculated using IRI-2016 and the GNSS-TEC data and between the TEC values calculated using upda-IRI-2016 and the GNSS-TEC data at different latitudes in 2015 and 2019, respectively. Here, IRI-TEC represents the TEC estimates calculated using the original IRI-2016 model and upda-IRI-TEC represents the TEC estimates calculated using the upda-IRI-2016 model.

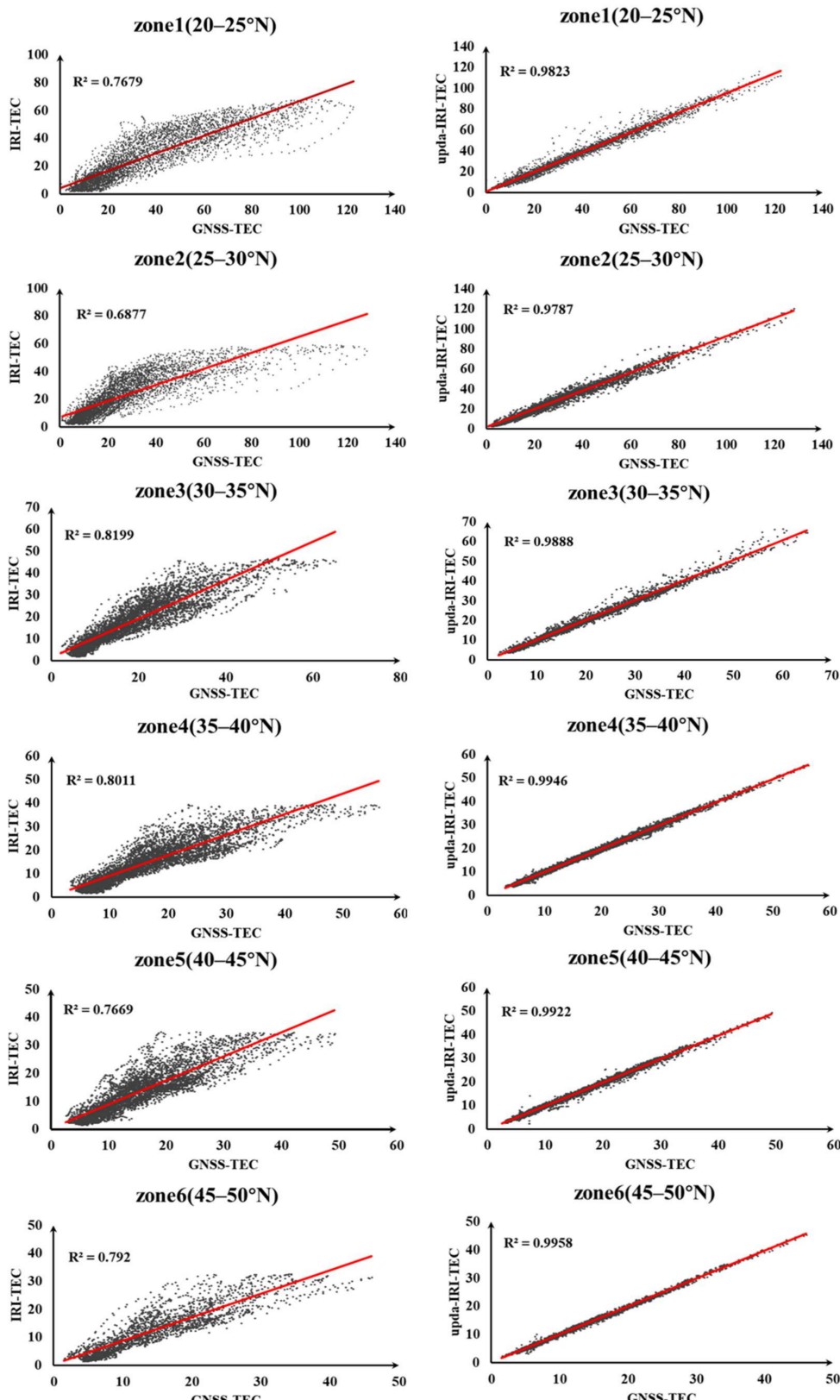

**Figure 9.** Linear regression diagrams between GNSS-TEC data from 2015 and the corresponding results of IRI-2016 and upda-IRI-2016 for 12 stations divided by latitude.

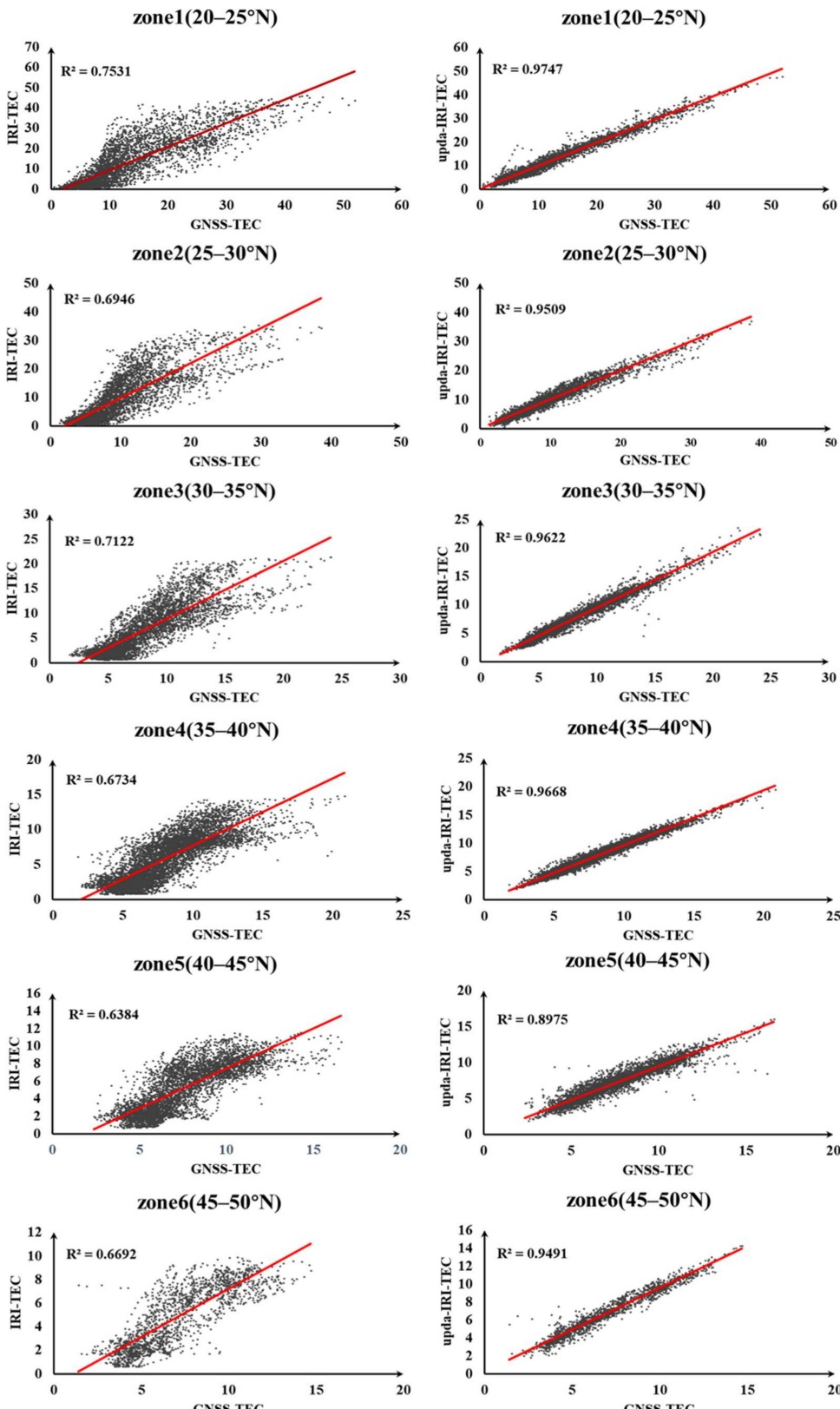

**Figure 10.** Linear regression diagrams between GNSS-TEC data from 2019 and the corresponding results of IRI-2016 and upda-IRI-2016 for 12 stations divided by latitude.

As seen from the linear regression diagrams, there is a significant correlation between upda-IRI-TEC and GNSS-TEC in 2015 and 2019. This correlation is above 0.90, especially for 2015, when it reaches above 0.97, and at middle-to-high latitudes, it is even higher than

0.99. In contrast, the correlation between the TEC values predicted by the original IRI-2016 model and GNSS-TEC is between 0.6 and 0.8.

Furthermore, for 2015 and 2019, the overall correlations between the TEC estimate calculated by IRI-2016 and the GNSS-TEC data from the 12 stations are 0.770 and 0.738, respectively, whereas those between the TEC estimates calculated by upda-IRI-2016 and the GNSS-TEC data are 0.986 and 0.966, respectively. The correlations between the TEC estimates calculated by upda-IRI-2016 and the GNSS-TEC data in years of both high and low solar activity are more than 0.2 higher than those achieved with the original IRI-2016 model.

## 4. Conclusions

In this article, high-precision TEC data obtained from 46 GNSS reference stations with a relatively uniform regional distribution over China were used to improve the IRI-2016 model, and the influence of the updated $IG_{12}$ index (IG-up) values on the TEC precision of the IRI-2016 model was discussed. Considering that the IG-up values vary greatly in time and space, 1-h, 2-h, and 4-h temporal interpolation schemes, a grid-based spatial interpolation scheme with spatial resolutions of 2.5° in latitude and 5° in longitude, and a spatial interpolation scheme based on the division into latitude zones over China were proposed, and their respective impacts on the accuracy of the TEC estimates calculated using the improved IRI-2016 model were demonstrated. Using high-precision TEC data obtained from 12 other GNSS reference stations in 2015 and 2019, the optimally integrated interpolation scheme for IG-up, with a 1-h temporal resolution and a 2.5° × 5° spatial resolution, was then evaluated in terms of its effectiveness in driving the IRI-2016 model to compute the TEC. The conclusions derived from the results are as follows:

1. Taking GNSS-TEC as a reference, we compared the ionospheric TEC estimates calculated using the IRI-2016 model driven by IG-up values obtained with different temporal interpolation schemes: ① The MAEs of the TEC estimates under the 1-h interpolation scheme for 2015 and 2019 are 0.5 TECu and 0.4 TECu, respectively; the MAE PIs relative to IRI-2016-TEC are 90.00% and 86.21%, respectively; the RMSEs are 0.6 TECu and 0.5 TECu, respectively; and the RMSE PIs relative to IRI-2016-TEC are 90.91% and 86.84%, respectively. ② The MAEs of the TEC estimates under the 2-h interpolation scheme for 2015 and 2019 are 0.6 TECu and 0.5 TECu, respectively; the MAE PIs relative to IRI-2016-TEC are 88.00% and 82.76%, respectively; the RMSEs are 0.9 TECu and 0.6 TECu, respectively; and the RMSE PIs relative to IRI-2016-TEC are 86.36% and 84.21%, respectively. ③ The MAEs of the TEC estimates under the 4-h interpolation scheme for 2015 and 2019 are 1.4 TECu and 0.8 TECu, respectively; the MAE PIs relative to IRI-2016-TEC are 72.0% and 72.41%, respectively; the RMSEs are 2.0 TECu and 1.1 TECu, respectively; and the RMSE PIs relative to IRI-2016-TEC are 69.70% and 71.05%, respectively. From these results, it can be seen that the 1-h interpolation scheme is the best.

2. Taking GNSS-TEC as a reference, we compared the ionospheric TEC estimates calculated using the IRI-2016 model driven by IG-up values obtained with different spatial interpolation schemes. According to the boxplots of the statistical results, the median differences (DTEC) between the TEC estimates calculated using the IRI-2016 model driven by IG-up values obtained via the grid interpolation scheme with a 2.5° × 5° spatial resolution and the GNSS-TEC data are closer to zero and more stable than those corresponding to the latitudinal-zone-averaging scheme. In addition, the upper and lower quartiles of the DTEC results of the grid interpolation scheme are more concentrated than those of the latitudinal-zone-averaging scheme, indicating that the former is the optimal space interpolation scheme for the IG-up values.

3. Taking GNSS-TEC as a reference, we evaluated the ionospheric TEC estimates calculated using the IRI-2016 model driven by IG-up values obtained with the optimally combined interpolation scheme: The MAE and RMSE for 2015 are 1.2 TECu and 1.4 TECu, respectively; compared with those of the original IRI-2016 model (5.6 TECu

and 6.6 TECu), the PIs are 78.57% and 78.79%, respectively. The MAE and RMSE for 2019 are 0.7 TECu and 0.8 TECu, respectively; compared with those of IRI-2016 (3.1 TECu and 3.5 TECu), the PIs are 77.42% and 77.14%, respectively. The correlations of linear regression with the GNSS-TEC data reach 0.986 and 0.966 for 2015 and 2019, respectively, being more than 0.2 higher than the corresponding correlations of IRI-2016-TEC with GNSS-TEC (0.770 and 0.738). In addition, overall, the TEC estimates calculated using the IRI-2016 model driven by IG-up values obtained with the comprehensive interpolation scheme show obvious improvements in years of both high and low solar activity, although the improvement effect in a year of high solar activity is better than that in a year of low solar activity.

Using the updated IG12 value to drive IRI-2016 greatly improves the accuracy of TEC calculated by IRI-2016. Therefore, based on the experimental results, we can provide an hourly IG-up grid map, with a spatial resolution of 2.5° in latitude and 5° in longitude. Users can extract the effective IG12 value of specific location and time according to the interpolation methods listed in this paper to ensure that in a region where there is no other more effective means (such as GNSS and ionosonde) to obtain high-precision ionospheric information, the IRI-2016 model can be used to reaches the corresponding accuracy requirements.

**Author Contributions:** Conceptualization, W.Z. and X.H.; methodology, X.H.; formal analysis, W.Z.; investigation, Y.Y.; resources, Z.L.; data curation, N.W.; writing—original draft preparation, W.Z.; writing—review and editing, W.Z. and X.H.; supervision, Y.Y. and Z.L. All authors have read and agreed to the published version of the manuscript.

**Funding:** This research is funded by the National Key Research & Development Program (No. 2017YFE0131400), National Natural Science Foundation of China (NO. 42074045). We also acknowledge the funding support provided by the State Key Laboratory of Geodesy and Earth's Dynamics (No. E025011003).

**Data Availability Statement:** Not applicable.

**Acknowledgments:** We acknowledge the Crustal Movement Observation Network of China (CMONOC) and Beijing Fangshan Satellite Laser Ranging National Observation and Research Station for providing access to GNSS data, The IRI-2016 Fortran source code can be downloaded from the IRI official website (http://www.irimodel.org, accessed in 2019). We acknowledge the use of data from the Chinese Meridian Project.

**Conflicts of Interest:** The authors declare no conflict of interest.

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
