# Peer review of "Algorithm Research Using GNSS-TEC Data to Calibrate TEC Calculated by the IRI-2016 Model over China"

_remotesensing, doi:10.3390/rs13194002_

Round 1

Reviewer 1 Report

This paper utilized GNSS-TEC data to modify the IRI model to let it better represent the real ionosphere over China. The paper discussed several interpolation methods (temporal, spatial and optimal combined) about coupling measured TEC into IRI, comparing the interpolated TEC and analyze the corresponding errors. The results are mainly in the geomagnetically quiet time and in both solar max and solar min. The results in this paper can be a potentially important contribution to the improvement IRI. Therefore, I recommend this paper to have a minor revision before it is suitable for publication

Line 12-13  I do not think that IRI is the most widely used model, please remove the ‘most’, this sentence is easy to make many scientist in ionosphere angry

Line 15-16  what is the ionospheric index?? The author shall give some brief explanation

Line 28-54 the author shall also provide some introduction of the ionosphere common phenomenon such as the Equatorial ionization anomaly, equatorial plasma bubbles, mid-latitude NmF2 enhancement and tongue of ionization

For EIA, the author can cite the following papers:

Cai, X., Burns, A. G., Wang, W., Qian, L., Liu, J., Solomon, S. C., et al. (2021). Observation of postsunset OI 135.6 nm radiance enhancement over South America by the GOLD mission. Journal of Geophysical Research: Space Physics, 126, e2020JA028108. https://doi.org/10.1029/2020JA028108

Laskar, F. I., Eastes, R. W., Martinis, C. R., Daniell, R. E., Pedatella, N. M., Burns, A. G., et al. (2020). Early morning equatorial ionization anomaly from GOLD observations. Journal of Geophysical Research: Space Physics, 125, e2019JA027487. https://doi.org/10.1029/2019JA027487

For bubbles, the author can cite

Karan, D. K., Daniell, R. E., England, S. L., Martinis, C. R., Eastes, R. W., Burns, A. G., & McClintock, W. E.(2020). First zonal drift velocity measurement of equatorial plasma bubbles (EPBs) from a geostationary orbit using GOLD data. Journal of Geophysical Research: Space Physics, 125, e2020JA028173. https://doi.org/10.1029/2020JA028173 

Martinis, C., Daniell, R., Eastes, R., Norrell, J., Smith, J., Klenzing, J., et al. (2021). Longitudinal variation of postsunset plasma depletions from the global-scale observations of the limb and disk (GOLD) mission. Journal of Geophysical Research: Space Physics, 126, e2020JA028510. https://doi.org/10.1029/2020JA028510

For tongue of ionization, the author can cite

Liu, J., W. Wang, A. Burns, S. C. Solomon, S. Zhang, Y. Zhang, and C. Huang (2016), Relative importance of horizontal and vertical transports to the formation of ionospheric storm-enhanced density and polar tongue of ionization, J. Geophys. Res. Space Physics, 121, 8121–8133, doi:10.1002/2016JA022882.

Line 232-238 the author first argued that the ionosphere TEC is greatly affected by latitude and solar activity, then they said that 6 stations are chosen, but is there any correlation or reason to choose these 6 stations?? Are there any special??

Furthermore, in section 3.1, the title is the comparison in different time intervals, but why there is no post-sunset comparison? As is known to us, the post-sunset ionosphere is also pretty important, but relatively poorly known compared with daytime variations

Author Response

Thanks for the comment.

Reviewer 2 Report

GNSS-TEC is used for adjustment of IRI model and the improvement is clearly shown. The overall results seem reasonable.

Author Response

Thank you very much for your valuable comments and support on this article!

Reviewer 3 Report

The analysis presented is of good quality but the conclusions are too descriptive and don't explain the relevance of the work. Is it really useful and why?  The fact that a short term parametrization works better than a long term average is not surprising, since we have a relatively low interval of ionospheric variation. What calls my attention is figure 2. The obtained values are very unstable meaning that the data is not enough to fit the underlying model. I would expect a simple polynomial fit to also give a "good" match.  As such the authors should explain why the need for such a complex model. The fact that 4 hours fits are still good is probably the most interesting result and points to the relevance of the model. Are those more stable? I would like to see what the authors obtain with 24 hour intervals. Stability of daily fits should also be tested.

Author Response

Thanks for the comments.

Round 2

Reviewer 3 Report

I am glad the authors followed my suggestions. The scope of the paper is now clear.